# A unique inhibitor conformation selectively targets the DNA polymerase PolC of Gram-positive priority pathogens

Mia Urem [1,2], Annemieke H. Friggen[1], Nina Musch [1], Michael H. Silverman [3], Christopher J. Swain [3], Michael R. Barbachyn[4], Lawrence I. Mortin [3], Xiang Yu[3], Robert J. DeLuccia [3], Meindert H. Lamers [5,6] ✉ & Wiep Klaas Smits [1,2] ✉

Infections with antimicrobial resistant pathogens are a major threat to human health. Inhibitors of the replicative polymerase PolC are a promising novel class of antimicrobials against Gram-positive pathogens, but the structural basis for their activity remains unknown. The first-in-class PolC-targeting antimicrobial, ibezapolstat, is a guanine analogue in late-stage clinical development for the treatment of *Clostridioides difficile* infections, and related inhibitors are being developed for systemic treatment of infections with methicillin-resistant *Staphylococcus aureus* (MRSA) and vancomycin-resistant enterococci (VRE). Here, we present the cryo-electron microscopy structures of *Enterococcus faecium* PolC bound to DNA and in complex with ibezapolstat or the previously-undescribed inhibitor ACX-801. Both inhibitors form base-pairing interactions with the DNA in the active site, thereby competing with incoming dGTP nucleotides. We identify a crucial susceptibility determinant in PolC that is conserved in other organisms, such as *C. difficile*. This is explained by an unusual non-planar conformation of the inhibitors that induce a binding pocket in PolC. By combining structural, biochemical, bioinformatic and genetic analyses, this work lays the foundation for the rational development of an innovative class of antimicrobials against Gram-positive priority pathogens.

The threat of antibiotic-resistant pathogens[1–3] requires the urgent development of novel antibiotics targeting unexploited biological processes that are essential for bacterial survival, such as DNA replication[4]. Bacterial replicative DNA polymerases, which belong to the prokaryote-specific C-family of polymerases, are desirable targets as they are structurally distinct from their eukaryotic counterparts[5–8]. C-family polymerases share the classic right-hand structural motif of DNA polymerases, where the thumb and finger domains position the DNA and incoming nucleotide (dNTP), while the palm domain

catalyses nucleotide incorporation complementary to the template DNA strand[9]. The C-family of polymerases can be further divided into PolC- (Pol IIIC) and DnaE-type polymerases[10,11]. In low GC-content Gram-positive bacteria (Bacillota, formerly Firmicutes) − which include priority pathogens like *Staphylococcus, Streptococcus, Clostridioides* and *Enterococcus* − PolC functions as the primary, processive enzyme responsible for the bulk of DNA synthesis whereas a DnaE homologue is thought to contribute to lagging strand DNA synthesis[12,13]. In contrast, Gram-negative bacteria and high GC-content

[1]Leiden University Center of Infectious Diseases (LUCID), Leiden University Medical Center, Leiden, The Netherlands. [2]Centre for Microbial Cell Biology, Leiden, The Netherlands. [3]Acurx Pharmaceuticals, Inc., Staten Island, NY, USA. [4]Calvin University, Grand Rapids, MI, USA. [5]Department of Cell and Chemical Biology, Leiden University Medical Center, Leiden, The Netherlands. [6]NeCEN—Netherlands Centre For Electron Nanoscopy, Leiden, The Netherlands. ✉e-mail: m.h.lamers@lumc.nl; w.k.smits@lumc.nl

Gram-positives have no PolC homologue, but instead employ a DnaE-type polymerase as the main replicative DNA polymerase. Structural differences between DnaE and PolC present opportunities for selective targeting[6]. Tailored inhibition of PolC thus offers a promising approach to combat Gram-positive pathogens while minimizing effects on organisms that lack PolC[14].

Ibezapolstat (IBZ; $N^2$-(3,4-dichlorobenzyl)-7-(2-[1-morpholinyl] ethyl)guanine; MorE-DCBG; formerly ACX-362E), developed by Acurx Pharmaceuticals, Inc., is a first-in-class $N^2$-substituted guanine antibiotic targeting the PolC polymerase and is in preparation for Phase 3 clinical trials[15–20]. It was developed for the treatment of *Clostridioides difficile* infection (CDI), the most common cause of healthcare-associated diarrhoea[21]. CDI has a high burden, in part due to frequent recurrences after initial clinical cure[21,22]. IBZ demonstrates activity against diverse isolates of *Clostridioides difficile*, including those with reduced susceptibility towards clinically used antimicrobials, and no pre-existing resistance to IBZ has been reported[23–25]. Importantly, for patients cured of CDI through treatment with IBZ during Phase 2 clinical trials, no recurrence was observed during the post-treatment study period[17,18]. The guanine-base moiety of IBZ (Fig. 1a) suggests that it acts as a dGTP-competitor, trapping PolC in a reversible-inactive state and stalling replication forks[16,23,26,27]. Notably, aromatic side-groups (3,4-dichlorobenzyl in IBZ) are believed to be important for specific inhibition of PolC and thereby Gram-positive bacteria[26,28,29]. However, no structure of PolC in complex with an inhibitor is available and the structural basis for the mode of action therefore remains unknown.

Here, we present the cryo-electron microscopy (cryo-EM) structure of PolC from vancomycin-resistant *Enterococcus faecium* (VRE; ATCC 700221), as a representative model for the replicative polymerase of Gram-positive bacteria, in complex with IBZ and a previously undescribed PolC inhibitor[30], ACX-801. Our structures reveal that the guanine nucleobase or its analogue base-pairs with the complementary dCMP in the DNA template strand, while the $N^2$-linked aromatic moiety is positioned at a ~ 90° angle from the nucleobase and inserted in an inhibitor-induced binding pocket that includes four aromatic residues. Cellular and biochemical analysis shows that one of the four aromatic residues, phenylalanine 1276 (F1258 in *C. difficile*), is a hotspot for mutations that drive resistance against IBZ and ACX-801. With this work, we lay the foundation for rationally developing improved inhibitors for the treatment of infections caused by Gram-positive priority pathogens.

## Results

### Inhibitors of PolC DNA polymerases

Due to its low solubility[18,19], IBZ is poorly absorbed and consequently only active in the gastrointestinal tract. In a laboratory setting, however, IBZ also effectively inhibits other clinically relevant Gram-positive pathogens such as vancomycin-resistant *E. faecium* (VRE), methicillin-resistant *Staphylococcus aureus* (MRSA) and penicillin-resistant *Streptococcus pneumoniae* (PRSP) (Table 1)[23], suggesting broader applicability of this class of antimicrobials. In search of more soluble PolC inhibitors, a series of IBZ derivatives were created through variation of the groups at the R1, R2 and R3 positions as exemplified for the

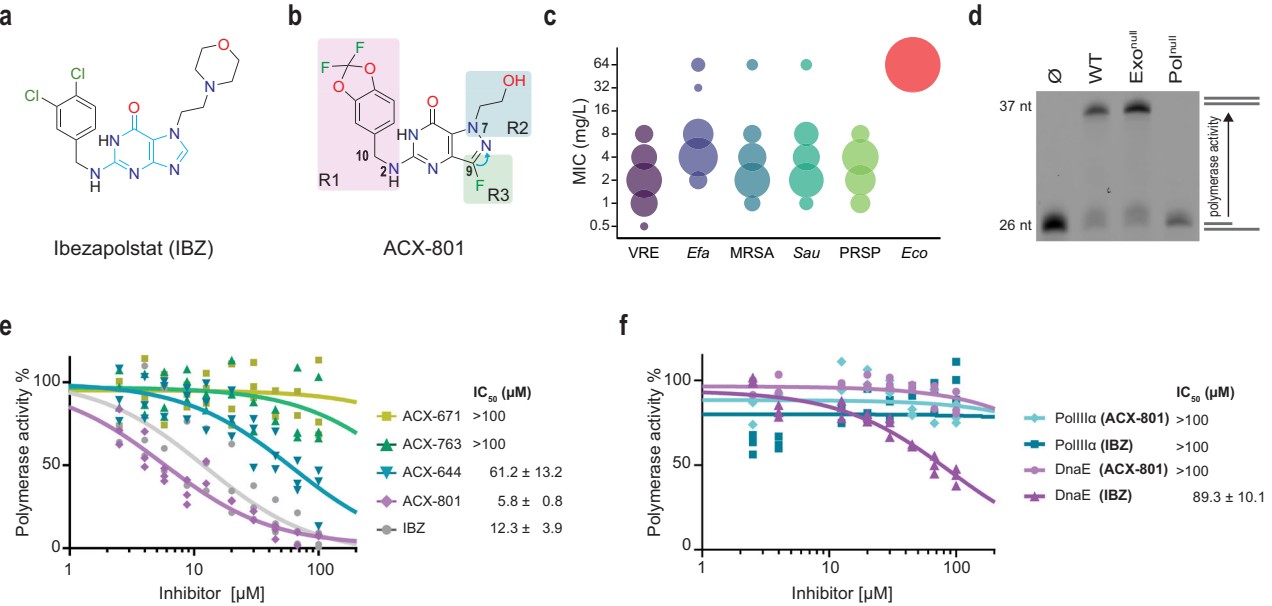

**Fig. 1 | PolC inhibitors selectively inhibit the replicative polymerase of Gram-positive bacteria. a** Structure of ibezapolstat (IBZ) with the guanine nucleobase moiety in light blue. **b** Structure of ACX-801 with the position of R1 ($N^2$-subtitution), R2 and R3 marked in coloured squares, and the blue arrow indicating displacement of the ring nitrogen from position 9 to 8 in the ACX scaffold compared to guanine and IBZ. **c** MIC distribution of 46 compounds from the ACX library for different bacterial species. Only MIC values below 16 mg/L for VRE were used. Values ≥ 64 mg/L are grouped together. Abbreviations: VRE (vancomycin-resistant *E. faecium*), *Efa* (*E. faecalis*), MRSA (methicillin-resistant *S. aureus*), *Sau* (susceptible *S. aureus*), PRSP (penicillin-resistant *S. pneumoniae*), and *Eco* (*E. coli*). The size of the circles corresponds the number of compounds, from 1 (smallest) to 46 (largest). **d** Gel-based primer extension assay showing DNA polymerase activity of *E. faecium* wild-type (WT) and exonuclease-inactivated PolC (Exo[null]; D431A + E433A), but not for polymerase-inactivated PolC (Pol[null]; D972A + D974A). Schematic of non-extended

primer:template DNA substrate is shown at the bottom right and fully extended primer:template above. A complementary exonuclease assay is shown in Supplementary Fig. 1a. A representative gel of multiple runs with reproducible results is shown. **e** Real-time assay (Supplementary Fig. 1b, c) measuring inhibition of polymerase activity by IBZ and 4 representative compounds from the ACX library using exonuclease-inactivated *E. faecium* PolC, with derived IC$_{50}$ values. Individual data points for each replicate ($n = 3$) are shown. A normalized dose-response (three parameter) fit was used to determine IC$_{50}$ values and, where an IC$_{50}$ could be determined ( < 100 µM), the standard error of the mean ($n = 3$) is given. **f** Real-time assay measuring susceptibility of DnaE-type polymerases *E. coli* Pol IIIα and *E. faecium* DnaE to IBZ and ACX-801. Individual data points for each replicate ($n = 3$) are shown. A normalized dose-response (three parameter) fit was used to determine IC$_{50}$ values and, where an IC$_{50}$ could be determined ( < 100 µM), the standard error of the mean ($n = 3$) is given.

**Table 1 | Antimicrobial specificity of nucleobase analogues against Gram-positive priority pathogens**

| Bacterial strain | Origin | Measured MIC (mg/L) | | | | | |
|---|---|---|---|---|---|---|---|
| PolC (Gram-positive) | | IBZ | ACX-801 | VAN | LZD | DAP | OMC |
| *E. faecium* VRE | ATCC 700221 | 2 | 1 | >64 | 4 | 4 | 0.12 |
| *E. faecalis* | ATCC 29212 | 8 | 2 | 4 | 2 | 8 | 0.25 |
| *S. aureus* MRSA | NRS 384 | 8 | 2 | 1 | 2 | ND | ND |
| *S. aureus* | ATCC 29213 | 8 | 2 | 1 | 4 | 1 | 0.5 |
| *S. pneumoniae* PRSP | ATCC 49619 | 2 | 4 | 0.25 | 1 | 0.5 | ≤0.06 |
| DnaE (Gram-negative) | | IBZ | ACX-801 | VAN | LZD | DAP | OMC |
| *E. coli* | ATCC 25922 | >64 | >64 | >64 | >64 | >64 | 1 |

The minimum inhibitory concentrations (MICs) of ibezapolstat (IBZ) and ACX-801 were compared to vancomycin (VAN), linezolid (LZD), daptomycin (DAP) and omadacycline (OMC) against a panel of Gram-positive priority pathogens, including vancomycin-resistant *Enterococcus faecium* (VRE), methicillin-resistant *Staphylococcus aureus* (MRSA), and penicillin-resistant *Streptococcus pneumoniae* (PRSP). All MIC experiments were performed as a single replicate; data from NRS384 is from an independent experiment that only included VAN and LZD as comparator.
*ND* not determined.

compound ACX-801 (Fig. 1b). For the latter group, the ring nitrogen at position 9 was moved to position 8. These changes were shown to improve systemic absorption following oral dosing as well as pharmacokinetics following intravenous dosing in rodents, key properties for development of a novel systemic antibiotic. Minimum inhibitory concentrations (MICs) were measured against five Gram-positive bacteria and one Gram-negative bacterium (Fig. 1c, Table 1). Overall, the ACX library demonstrated selective inhibition of Gram-positive bacteria, with some compounds (including ACX-801) demonstrating potent activity against drug-resistant priority pathogens (VRE, MRSA and PRSP), while no activity was observed against the Gram-negative bacterium *E. coli*.

A selection of 48 ACX candidates with MICs <16 mg/L against VRE, MRSA and/or PRSP was evaluated in an in vitro DNA polymerase assay[31,32] using purified PolC from *E. faecium* (ATCC 700221). We validated the activity of the purified enzyme with a gel-based primer extension assay using the wild-type version, an exonuclease-inactivated version (Exo$^{null}$, PolC$^{D431A+E433A}$) and a polymerase catalytically dead mutant (Pol$^{null}$, PolC$^{D972A+D974A}$) (Fig. 1d, Supplementary Fig. 1a). Both the wild-type and the exonuclease-inactivated variant show robust DNA polymerase activity, while the polymerase-inactivated mutant shows no activity. We employed the exonuclease-inactivated protein in subsequent real-time DNA polymerase assays to measure the effect of different ACX compounds on the activity of PolC, without the potential interference of exonuclease activity (Supplementary Fig. 1b, c). In this assay, IBZ and ACX-801 exhibited the strongest inhibition with an inhibitory concentration at which activity is 50% (IC$_{50}$) of 12.3 and 5.8 μM, respectively, while most other compounds showed IC$_{50}$ values of 60 μM or higher (Fig. 1e). ACX-801 (5-(((2,2-difluorobenzo[d][1,3]dioxol-5-yl)methyl)amino)-3-fluoro-1-(2-hydroxyethyl)-1,6-dihydro-7H-pyrazolo[4,3-d]pyrimidin-7-one) differs from IBZ in the appended aromatic moiety at the $N^2$ position (R1), where the 3,4-dichlorobenzyl in IBZ is substituted with a larger and more complex (2,2-difluoro-1,3-benzodioxol-5-yl)methyl moiety. Additionally, the morpholine group at the R2 position is replaced by a hydroxyl group in ACX-801 and a fluorine is introduced at the R3 position (Fig. 1a, b). Compared to IBZ, ACX-801 demonstrates a 2- to 4-fold increased potency against Gram-positive bacteria, including MRSA and VRE (Table 1).

ACX-801 shows selective inhibition of *E. faecium* PolC, with no significant inhibition observed against DnaE-type DNA polymerases: Pol IIIα from *E. coli*[33,34] and DnaE of *E. faecium* (Fig. 1f). In contrast, IBZ showed some inhibition towards *E. faecium* DnaE in vitro, though ~seven-fold lower than PolC (IC$_{50}$: 89.3 μM). This suggests that the larger moiety of ACX-801 cannot be accommodated by DnaE-type DNA polymerases, though we cannot rule out that other factors contribute to the reduced inhibition.

## Cryo-EM structures reveal a unique inhibitor conformation

We employed single-particle cryo-EM to determine an atomic model of PolC in complex with a DNA substrate and either IBZ or ACX-801 (Fig. 2). To prevent the degradation of the DNA substrate, we again used the exonuclease-inactivated PolC (Exo$^{null}$, PolC$^{D431A+E433A}$). In the cryo-EM data of both the IBZ and ACX-801 samples, two distinct structural states were discerned: an 'apo state' in which no double-stranded DNA (dsDNA) substrate (nor inhibitor) is observed in the cryo-EM map (Fig. 2a) and a 'polymerase state' in which the dsDNA substrate is positioned in the polymerase active site together with the inhibitor (shown in Fig. 2b for ACX-801). The three structures (PDB-9QPC, 9QRN and 9QRL) were resolved to a resolution of 2.8–3.2 Å (Supplementary Table 1 and Supplementary Figs. 2, 3). Due to the lower resolution of the IBZ-bound structure, only 15 residues of the OB domain (consisting of 110 residues) and 175 residues of the exonuclease domain (consisting of 211 residues) could be modelled in this map. Both domains are located at the periphery of the polymerase. In comparison, in the higher-resolution ACX-801-bound structure, only 12 residues were omitted in the OB domain and 2 residues in the exonuclease domain. The overall 3D fold of *E. faecium* PolC resembles the structure of other C-family polymerases[10,34,35] and follows the same predicted domain organization as PolC-type polymerases (Fig. 2c), but is most similar to PolC of the non-pathogenic bacterium *Geobacillus kaustophilus*[6] (Supplementary Fig. 4), that was determined by X-ray crystallography. *G. kaustophilus* PolC is the only experimentally determined PolC structure to date, but lacks the exonuclease domain, which was removed for crystallization purposes, and additionally does not contain any inhibitor[6]. In our structures of full-length *E. faecium* PolC, the exonuclease domain is resolved and localised between the third and fourth strand of the Polymerase and Histidinol Phosphatase (PHP) domain (Fig. 2a–c, Supplementary Fig. 5). In both the apo and polymerase state, a three-nucleotide single-stranded DNA (ssDNA) oligo of unknown origin is positioned in the exonuclease active site in a manner similar to other DnaQ-like exonucleases found in *E. coli* Pol I and *E. coli* Pol III (Fig. 2a, b, Supplementary Fig. 5). However, in *E. coli* PolIIIα, the exonuclease domain, which is the separate subunit ε, is positioned on top of the thumb domain, while in PolC it is located below the thumb domain. When viewed along the axis of the DNA, this is a rotation of ~70° (Supplementary Fig. 5).

In the polymerase state, the dsDNA is held between the thumb, palm, and fingers domain, analogous to structures of other C-type DNA polymerases (Fig. 2c, Supplementary Fig. 4). The inhibitors IBZ and ACX-801 are only observed in the polymerase state structures, where several residues interact with the R1 and R2 groups (Fig. 2d, e), and the nucleobase moiety of the inhibitor can stack onto the 3′ terminal base of the primer strand (Supplementary Figs. 2e and 3e) while base-pairing with the dCMP in the template strand (Fig. 2f). The nucleobase

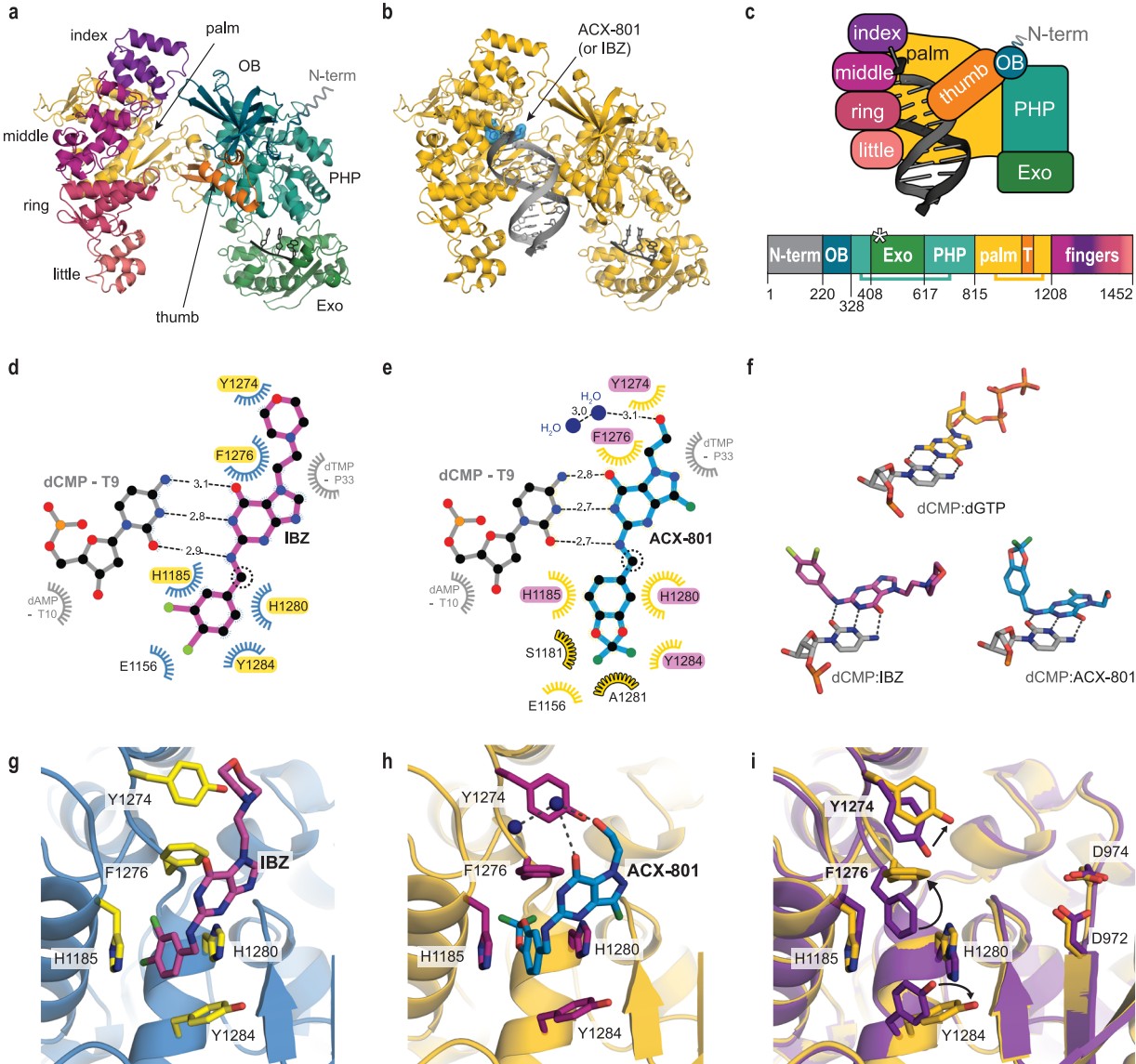

**Fig. 2 | Cryo-EM structures of *E. faecium* PolC with IBZ and ACX-801. a** Apo structure (PDB-9QRN) of exonuclease-inactivated *E. faecium* PolC with a 3-nucleotide ssDNA bound to the exonuclease domain (Exo). Other domains are labelled OB (oligonucleotide/oligosaccharide-binding) and PHP (polymerase and histidinol phosphate) as in 2c. The flexible N-terminal domain (N-term), not resolved in the density map and structure, is indicated in grey. **b** Structure of exonuclease-inactivated *E. faecium* PolC (yellow) bound to DNA (grey) and ACX-801 (blue) (PDB-9QPC). **c** Schematic representations of the domains of *E. faecium* PolC. The PHP domain is interrupted by the Exo domain, and the palm domain is split by the thumb (T) subdomain. The position of the catalytic residues of the Exo domain (D431 and E433) are indicated with an asterisk. **d** Ligand interaction map of IBZ as derived from the structure (PDB-9QRL). An alternative map containing further details is provided in Supplementary Fig. 6. Conserved interacting residues are highlighted. **e** Ligand interaction map of ACX-801 as derived from the structure (PDB-9QPC). An alternative map is provided in Supplementary Fig. 6. Residues uniquely identified as interacting with ACX-801 are indicated with a stroke. Conserved interacting residues are highlighted. **f** Base-pairing (represented by dashed lines) between the dCMP (grey) and dGTP (yellow, from PDB-3F2C), IBZ (pink, PDB-9QRL) and ACX-801 (blue, PDB-9QPC). Stick representations are coloured by atom but with different backbone colours. **g** Close-up of the binding pocket in the IBZ-bound PolC structure (PDB-9QRL). PolC is shown in blue with specific residues in yellow and IBZ as sticks with a purple backbone. **h** Close-up of the binding pocket in the ACX-801-bound PolC structure (PDB-9QPC). PolC is shown in yellow with specific residues in pink and ACX-801 in blue. Water is represented as a blue spheres and the dashed lines indicate interactions with residue Y1274. **i** Displacement of residues in the ACX-801-bound structure (yellow; PDB-9QPC) compared to the ligand-free, apo structure (purple; PDB-9QRN). Polymerase catalytic residues (D972 and D974) are annotated. The arrows highlight the rotation and displacement of F1276, and minor displacement of Y1274 and Y1284 to accommodate the inhibitor.

moiety of the inhibitor occupies the same position as the nucleobase of the incoming dGTP nucleotide in the *G. kaustophilus* PolC structure[6], by analogy showing that the inhibitor acts by blocking access for incoming nucleotides (Fig. 2f, Supplementary Fig. 4).

Notably, both IBZ and ACX-801 adopt a distinct non-planar conformation where the aromatic moiety at the R1 position (i.e. 3,4-dichlorobenzyl in IBZ or (2,2-difluoro-1,3-benzodioxol-5-yl)methyl in ACX-801) is at a ~ 90⁰ angle compared to the plane of the nucleobase

(Fig. 2f, Supplementary Figs. 2–4). These R1-moieties are inserted into a shallow, hydrophobic pocket that includes four aromatic residues: histidine 1185, phenylalanine 1276, histidine 1280, and tyrosine 1284 (H1185, F1276, H1280, and Y1284, respectively) (Fig. 2g, h, Supplementary Figs. 2, 3, and 6). This pocket is not observed in the apo state structure or in the structure of *G. kaustophilus* PolC (Supplementary Fig. 4). Rather, in these structures, the inhibitor binding pocket is blocked by F1276, which moves outwards upon binding of the inhibitor

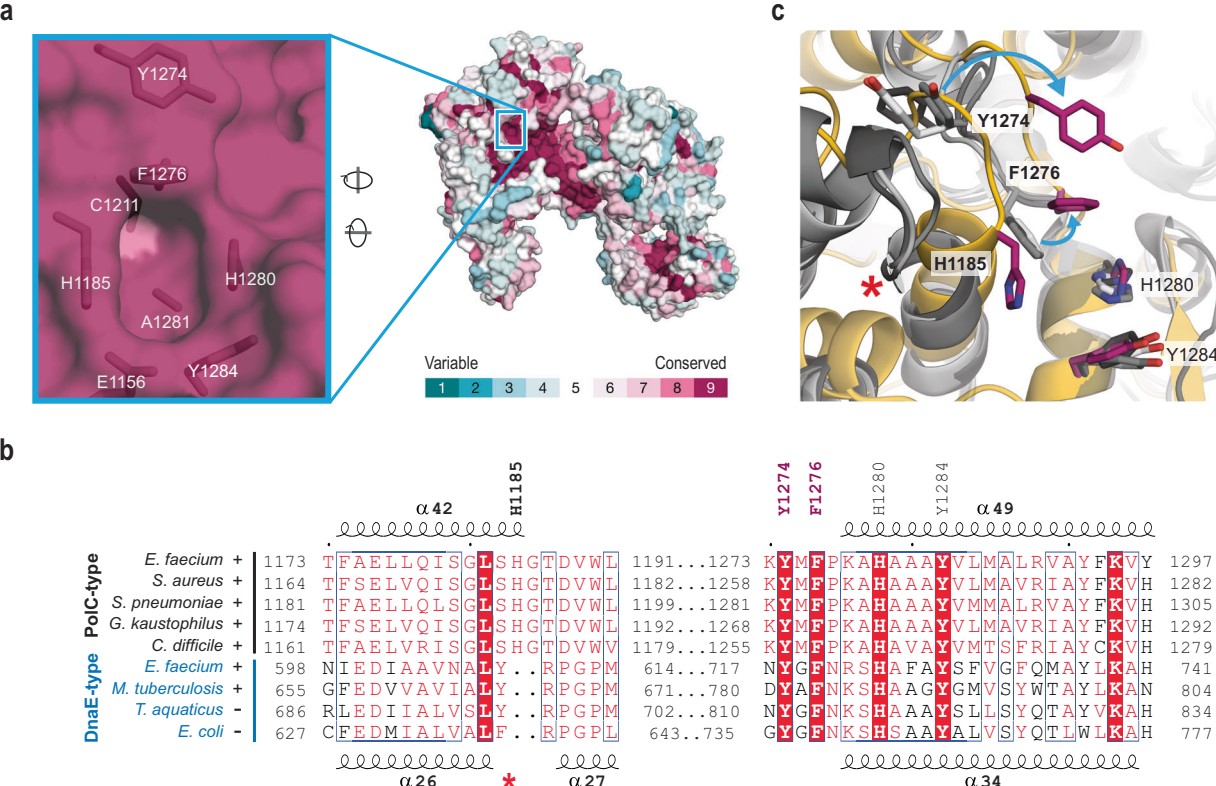

**Fig. 3 | Structural explanation for selectivity of PolC inhibitors. a** Sequence conservation plotted on the surface of PolC (PDB-9QPC), showing the highest conservation in the DNA binding cleft and exonuclease active site. The inset on the left shows the conservation of PolC sequences plotted on an enlargement of the surface of the inhibitor binding pocket, with relevant amino acid residues indicated. **b** Structure-based sequence alignment of C-family sequences from Gram-positive and Gram-negative bacteria. Species names are coloured according to PolC-type (black) and DnaE-type (blue) polymerase sequences, with a + or − indicating Gram-positive or -negative bacteria, respectively. The secondary structures (coils representing helices numbered according to PDBs) are shown above for *E. faecium* PolC (PDB-9QPC) and below for *E. coli* PolIIIα (PDB-5M1S). The residues

that are part of the inhibitor-binding pocket of *E. faecium* PolC are indicated, with residues that are displaced in the inhibitor-bound conformation in pink. The red asterisk (*) marks the truncated helix in DnaE-type polymerases. **c** Superposition of *E. faecium* PolC in ACX-801-bound form (in yellow) and three DnaE-type polymerases (in three tones of grey): *E. coli* PolIIIα (PDB-5M1S), *M. tuberculosis* DnaE1 (PDB-7PU7), and *Thermus aquaticus* Pol IIIα (PDB-3E0D). The two arrows indicate the movement of residues required to create a full inhibitor binding pocket. The red asterisk (*) marks the end of the helix in DnaE-type polymerases; this helix is extended by half a turn in PolC-type polymerases and positions the fourth aromatic residue of the PolC inhibitor binding pocket, which has no structural equivalent in DnaEs. Numbering of residues is based on *E. faecium* PolC.

(Fig. 2i, Supplementary Movie). In the binding pocket, H1185 and H1280 are positioned parallel to the R1 moiety of IBZ and ACX-801 and make π-π stacking interactions. F1276 and Y1284 are positioned perpendicular to the ring of the R1 moiety and make contact via edge-to-face interactions with the aromatic ring hydrogens of R1. F1276 also makes an additional edge-to-face interaction with the nucleobase of the inhibitor. A further contact is made outside the binding pocket with tyrosine 1274 (Y1274). This residue stacks onto the protonated morpholine group at the R2 position of IBZ but engages in a hydrogen bond with a water molecule located above the nucleobase (distance 3.0 Å) rather than interacting with the hydroxyl group at the R2 position of ACX-801 (distance 4.2 Å) (Fig. 2d, e, Supplementary Fig. 2e, f).

The four aromatic residues that form part of the binding pocket (H1185, F1276, H1280, and Y1284) are strictly conserved in PolC DNA polymerases, including those of priority pathogens such as MRSA and PRSP (Fig. 3a, b). However, there is no structural equivalent of H1185 in the structures of *E. coli* Pol IIIα (PDB-5M1S), *Thermus aquaticus* Pol IIIα (PDB-7PU7) and *Mycobacterium tuberculosis* DnaE1 (PDB-3E0D)[5,34,36] as the helix containing this residue in PolC is shorter in the DnaE-type DNA polymerases (Fig. 3b, c). Furthermore, the equivalent of Y1274, that in PolC contacts or comes close to the R2 group, is positioned away from the inhibitor binding pocket in DnaE-type polymerases (Fig. 3c). These

differences may explain in part the higher specificity of these inhibitors against Bacillota over other groups of bacteria (Table 1), and the generally lower inhibition of DnaE-type polymerases in vitro (Fig. 1f).

## A key phenylalanine is essential for activity of PolC inhibitors

To gauge the potential resistance development against ACX compounds, we exposed *E. faecium* ATCC 700221 (VRE) to various early lead compounds from the ACX library (ACX-641, ACX-671, ACX-728, ACX-763) at 2-16x the MIC. This led to the isolation of several mutants harbouring mutations in the *polC* gene (Table 2, BioProject PRJNA1190376). Notably, mutations in *polC* were mostly observed at ≥4x MIC. The frequency of reduced susceptibility in these single-step experiments at 4x MIC and 8x MIC was generally $<7.7 \times 10^{-8}$, similar to what has been reported for other PolC inhibitors[37], and no strains were recovered at 16x MIC (Supplementary Table 2). Strains carrying mutations that were selected on one ACX compound showed varying levels of cross-resistance to other ACX compounds, including IBZ and ACX-801, but not to linezolid, daptomycin or omadacycline (Table 2).

Particularly significant were mutations of F1276, the gatekeeper of the induced binding pocket. Based on the cryo-EM structures, the three mutations (F1276I, F1276L and F1276S) that were observed in the

**Table 2 | Reduced susceptible strains show cross-resistance against other PolC inhibitors but not to antimicrobials with a different mode of action**

| polC allele | | Strain | Selected on | | Measured MIC (mg/L) | | | | | | | | |
|---|---|---|---|---|---|---|---|---|---|---|---|---|---|
| | | | mg/L | ACX | IBZ | 801 | 641 | 671 | 728 | LZD | VAN | DAP | OMC |
| WT | | ATCC 700221 | – | – | 2 | 1 | 1 | 2 | 4 | 4 | >64 | 4 | 0.12 |
| A1281 | T | Efa040 | 4 | 671 | 8 | 32 | 8 | 16* | 8 | 4 | >64 | 4 | 0.12 |
| | | Efa036 | 2 | 641 | 16 | 16 | 8* | 16 | 8 | 4 | >64 | 4 | 0.12 |
| F1276 | I | Efa037 | 4 | 641 | >64 | >64 | 32* | >64 | 64 | 4 | >64 | 2 | ≤0.06 |
| | L | Efa030 | 16 | 728 | >64 | 64 | 64 | >64 | >64* | 4 | >64 | 2 | 0.12 |
| | S | Efa3 | 16 | 728 | 16 | 32 | 4 | 16 | >64* | 4 | >64 | 2 | ≤0.06 |

Selected *E. faecium* strains were assessed for susceptibility to PolC inhibitors. Minimal inhibitory concentration (MIC) values are indicated in the table. Asterisks indicate the antimicrobial that was initially used for mutant selection. For simplicity, 'ACX' from the ACX compounds has been omitted, such that ACX-801 is shown as 801. *VAN* vancomycin, *LZD* linezolid, *DAP* daptomycin, *OMC* omadacycline.

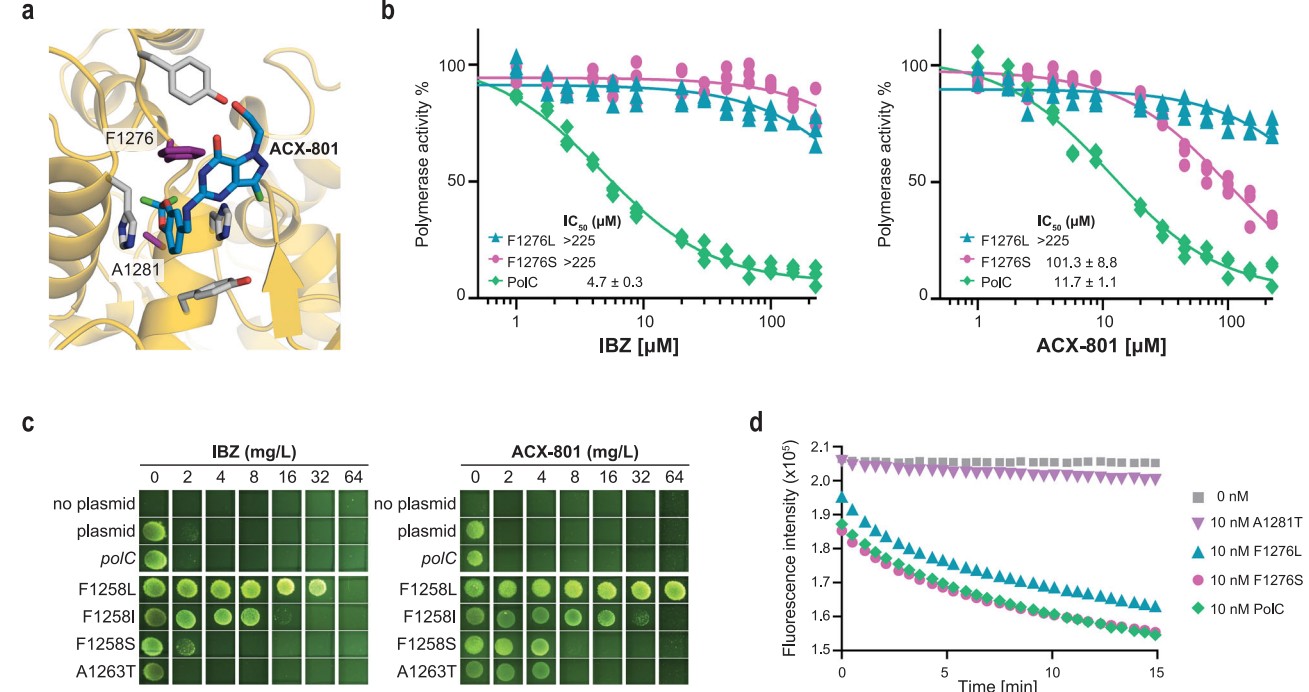

**Fig. 4 | A specific phenylalanine residue is a key susceptibility determinant for PolC inhibitors. a** Close-up of the inhibitor binding pocket in exonuclease-inactivated *E. faecium* PolC (PDB-9QPC) with two residues involved in resistance to PolC inhibitors (F1276 and A1281) highlighted in purple. ACX-801 is shown in blue and other residues that make up the binding pocket are shown in grey. **b** Polymerase activity inhibition of F1276 mutants compared to wild-type PolC protein by IBZ (left) and ACX-801 (right), with derived IC$_{50}$ values. All proteins were exonuclease-inactivated. Individual data points for each replicate (*n* = 3) are shown with a normalized dose-response (three parameter) fit and, where an IC$_{50}$ could be determined, the standard error of the mean is given. **c** Susceptibility of *C. difficile* carrying different plasmid-based *polC* alleles to IBZ and ACX-801. Cells were spotted onto BHI agar with increasing amounts of IBZ (left) or ACX-801 (right). Plasmids carrying the wild-type *polC* gene (*polC*), or mutant alleles *polC*[p.F1258L] (F1258L), *polC*[p.F1258I] (F1258I), *polC*[p.F1258S] (F1258S) or *polC*[p.A1263T] (A1263T) are shown. **d** DNA polymerase activity of different PolC variants in the absence of inhibitor. The PolC[A1281T] (A1281T) mutant shows an ~10-fold reduction in activity compared to wild-type PolC and two different F1276 variants. The activity is shown as an average of three replicates; the standard error is omitted as it is obscured by the size of the symbols.

resistance selection experiments likely lead to loss of the interactions with both the $N^2$ moiety and the nucleobase (Fig. 4a, Fig. 2d, e, g, h, Supplementary Fig. 6). Mutations analogous to mutations of F1276 in *E. faecium* PolC have been observed in different Gram-positive bacteria in response to 6-anilinouracil inhibitors HPUra (6-(p-hydroxyphenylazo)-uracil), ME-EMAU (2-methoxyethyl-6-(3'-ethyl-4'-methylanilino)uracil), and HB-EMAU (N3-hydroxybutyl 6-(3'-ethyl-4'-methylanilino) uracil)[37–39]. 6-anilinouracil nucleobase analogues also inhibit PolC; the aromatic 6-anilino group is located at an analogous position to the $N^2$ aromatic moieties in IBZ and ACX-801. This shows that F1276 is a general susceptibility determinant for nucleobase inhibitors of PolC beyond ACX-801 and IBZ.

We also purified *E. faecium* PolC[F1276L] and PolC[F1276S] to investigate the impact of the F1276 mutations on polymerase activity and inhibition by IBZ and ACX-801 in vitro. Both PolC variants retained polymerase activity comparable to or modestly lower than the wild-type enzyme in our assay (Fig. 4b). We note that in *S. aureus* the equivalent protein (PolC[F1261L]) showed a reduced rate of nucleotide incorporation in pre-steady-state kinetic experiments in the presence of clamp protein, but showed comparable rates to wild type in a steady-state gel-based polymerase assay[39]. The F1276L mutation increased the IC$_{50}$ in our assay by >48-fold and >19-fold for IBZ and ACX-801, respectively, whereas the F1276S mutation resulted in a similar >48-fold increase for IBZ and a > 8-fold increase in IC$_{50}$ for ACX-801. Together, these data

demonstrate the relevance of the F1276 position for susceptibility to PolC inhibitors both in vivo and in vitro.

As no mutations have been described to date that reduce the susceptibility of *C. difficile* to IBZ, we leveraged the information from *E. faecium* and introduced variant *polC* genes on a plasmid into the laboratory strain *C. difficile* 630Δ*erm*[40]. The wild-type strain has an MIC of 2 mg/L for both IBZ and ACX-801 under the conditions tested. Introduction of plasmids carrying *polC*-p.F1258L, *polC*-p.F1258I and *polC*-p.F1258S (encoding *C. difficile* PolC with mutations of F1258, the equivalent of F1276 in *E. faecium* PolC) increased the MIC 32-, 8- and 2-fold for IBZ and ≥32-, 32- and 4-fold for ACX-801, respectively (Fig. 4c). These data underscore that the gatekeeper phenylalanine is equally important for susceptibility to PolC inhibitors in *C. difficile*.

We also identified a previously undescribed resistance-associated *polC* mutation in *E. faecium*: an alanine-to-threonine substitution at position 1281 (A1281T) that raises the MIC up to 32-fold for ACX-801 and up to eightfold for IBZ (Table 2). A1281 is buried in the apo state but accessible in the induced pocket observed in the polymerase state structure (Figs. 2e, 3a), and is oriented at the top of the R1 moiety (Fig. 4a). In contrast to the F1276 mutations, the A1281T mutation strongly reduced the DNA polymerase activity of the purified protein in vitro (Fig. 4d), though the *E. faecium* strain harbouring the *polC*-p.A1281T mutation was viable. It is conceivable that in vivo the presence of other replisome subunits sufficiently compensates for the reduced polymerase activity of this mutant PolC. We also assessed the effect of introduction of the equivalent mutation (A1263T) on resistance in *C. difficile*. Introduction of a plasmid carrying *polC*-p.A1263T leads to a fourfold increase in the MIC for ACX-801, but not for IBZ (Fig. 4c). This suggests that the larger moiety of ACX-801 at the R1 position allows for the selection of resistance mutations further in the pocket that do not affect the interaction with IBZ. Our data suggest that the functional importance of the alanine residue is conserved across organisms, as also observed for the phenylalanine residue.

## Discussion

There is an urgent need for the development of antimicrobials with a novel mode of action to combat the global threat of Gram-positive antimicrobial resistance[41,42]. Chemically synthesized PolC inhibitors represent a new-to-nature class of compounds that demonstrate great promise for the treatment of infections caused by Gram-positive priority pathogens[29]. The development of the next generation of this class of antimicrobials requires insight into the structural determinants for inhibition and resistance. Here, we presented a model of PolC in complex with IBZ, an orally dosed non-absorbed drug in clinical development for the treatment of *C. difficile* infections, as well as the previously undescribed inhibitor, ACX-801, as a representative of PolC inhibitors with systemic antibacterial activity following oral and intravenous dosing in rodents.

The cryo-EM structure of IBZ, as well as ACX-801, bound to *E. faecium* PolC and DNA supports a model in which $N^2$-substituted nucleobase analogues adopt a distinct non-planar conformation leading to an induced fit that explains the relevance of the aromatic moiety for inhibition[26,28,29]. To our knowledge, the strong bend in this type of polymerase inhibitor molecule has not directly been observed to date. However, based on biochemical experiments with 6-amino- and 6-hydrazinouracil inhibitors of PolC, an "active conformation" of the inhibitors with the phenyl moiety in *cis* conformation compared to the pyrimidine moiety was hypothesized[43], consistent with our findings. The overall fold of the full length *E. faecium* PolC protein in complex with the inhibitors is highly similar to the published structure of the *G. kaustophilus* PolC catalytic domain in complex with dsDNA and dGTP[6], and models of full-length *E. faecium* PolC in the apo state as predicted by Alphafold[44,45]. However, modelling based on these protein structures does not reveal the interactions observed here as the structure of uracil-based PolC inhibitors in solid state is planar[43],

modelling of the next generation PolC inhibitors shows a similar extended conformation (Supplementary Fig. 7) and the active conformation is dependent on an induced fit.

Our results highlight a crucial role for F1276 as a susceptibility determinant. The phenylalanine residue is important for stabilizing and positing incoming nucleotides, helping to ensure proper alignment with the template strand during polymerization[46]. Mutation of this residue to leucine, isoleucine or serine results in up to 64-fold reduced susceptibility to the inhibitors. Though we find that introduction of different *polC* mutations in *C. difficile* can increase the MIC for PolC inhibitors, we have so far only obtained a single *C. difficile* strain carrying a F1258L mutation with reduced susceptibility to IBZ (selected on 8 mg/L; MIC = 64 mg/L) by one-step passage under laboratory conditions. The implications of the reduced susceptibility resulting from *polC* mutations for treatment is unclear; at least for IBZ, faecal levels reach >1000 µg/g stool[19], which likely exceeds the MIC by several orders of magnitude. Moreover, it can be envisaged that some resistance mutations may be associated with a fitness cost in vivo, as we observed strongly reduced polymerase activity for the A1281T mutation in vitro.

The similar conformation of ACX-801 and IBZ, together with high sequence conservation of the polymerase active site (and the residues that form the induced binding pocket in particular) suggests that the mechanism reported here for *E. faecium* (VRE) is a general mechanism for this type of inhibitor and will be similar in other organisms like *C. difficile*, *S. aureus* (MRSA) and *S. pneumoniae* (PRSP). Moreover, the data presented here strongly suggest that nucleobase analogues can be tailored to enhance potency and specificity while mitigating resistance development, paving the way for a next generation of targeted antimicrobial therapies against Gram-positive pathogens.

## Methods

### Reagents and materials

All chemicals were purchased from Sigma-Aldrich, unless indicated otherwise. IBZ and ACX compounds were provided by Acurx Pharmaceuticals, Inc. Chromatography columns were purchased from Sigma-Aldrich and Cytiva. For grid preparation, Quantifoil R0.6/1 holey carbon grids (Quantifoil) were used. DNA oligonucleotides were purchased from IDT or Sigma-Aldrich. 384-well Black Round Bottom plates (#4514) were obtained from Corning. SequaGel UreaGel 19:1 Denaturing Gel System was obtained from National Diagnostics.

### Alignment and analysis of PolC protein sequences

To compare PolC and DnaE sequences, MAFFT[47] and MultAlin[48] were used for multiple sequence alignment. PolC sequences were obtained for *E. faecium* (ATCC 700221; locus_tag: MIBBHLJO_01574), *S. aureus* (NRS 384; MRSA; accession: A6QGG4.1), *S. pneumoniae* (ATCC 49619; PRSP; accession: WP_050206261.1), *C. difficile* (accession: WP_003438301.1) and *G. kaustophilus* (as PDB-3F2B; accession: BAD75543.1).

To determine the sequence conservation of PolC polymerases, more than 221 additional PolC sequences (excluding any DnaE sequences) were obtained from the curated data set by Timinskas et al.[11]. These were combined with the PolC sequences listed above and expanded to include *E. faecalis* ATCC 29212 (vancomycin-susceptible; accession: WP_002381593.1), *Bacillus subtilis* str. 168 (accession: AQR81638.1), *S. aureus* ATCC 29213 (methicillin-susceptible; accession: A7X1P4.1), *Streptococcus agalactiae* (accession: WP_001292137.1) and *Streptococcus pyogenes* (accession: WP_011285722.1). The full set of PolC sequences was aligned with MAFFT[47], and the conservation scores of residues were determined and mapped on the ACX-801-bound PolC structure (PDB-9QPC) by ConSurf Web Server (accessed September and December 2024)[49,50].

For structure-based alignment, the regions around the polymerase active site of the different C-family polymerases were aligned

based on the secondary structure of *E. faecium* PolC (this work, PDB-9QPC) and *E. coli* K-12 Pol IIIα (PDB-5M1S). DnaE sequences were obtained for: *E. coli* K-12 (as PDB-5M1S; accession: 5M1S_A), *T. aquaticus* (as PDB-3E0D; accession: AAD44403.1) and *M. tuberculosis* (as PDB-7PU7; accession: WP_003407751.1). Experimentally derived structures were used for *G. kaustophilus* PolC (PDB-3F2B), *M. tuberculosis* DnaE1 (PDB-7PU7) and *T. aquaticus* DnaE (PDB-3E0D). Where no experimental structure was available, models with 100% identical sequences were obtained from the AlphaFold[44,45] database for *S. aureus* (AF-A6QGG4-F1-v4) and *S. pneumoniae* (AF-Q97SQ2-F1-v4) or generated using ColabFold v1.5.5[51] (AlphaFold2 using MMseqs2 with default settings) for *C. difficile*. ESPript 3.0[52] was then used to depict the structure-based alignment of PolC-type and DnaE-type polymerases, and sequence similarity was annotated based on the Risler scoring system.

### Broth microdilution-based susceptibility testing

Minimal inhibitory concentrations (MICs) were determined by WuXi Apptec, on behalf of Acurx Pharmaceuticals, Inc., for the Gram-positive pathogens: *Enterococcus faecium* ATCC 700221 (VRE; vancomycin-resistant), *Enterococcus faecalis* ATCC 29212, *Staphylococcus aureus* NRS384 (MRSA; methicillin-resistant), *Staphylococcus aureus* ATCC 29213 (MSSA; methicillin-sensitive), and *Streptococcus pneumoniae* ATCC 49619 (PRSP; penicillin-resistant). As a control, Gram-negative *Escherichia coli* ATCC 25922 was included. All strains were obtained from ATCC with the exception of *Staphylococcus aureus* NRS384, which was obtained from NARSA (Network on Antimicrobial Resistance in *Staphylococcus aureus*, now part of BEI resources). Bacteria were cultured according to the manuals provided by the suppliers and stored in 25% glycerol (final concentration) at −80 °C.

Bacterial strains were streaked out onto MHA plates (Mueller-Hinton II Agar; BD-211438) and incubated at 37 °C for 20 h except for *S. pneumoniae* and *Enterococci*, which were streaked on blood agar (Huankai-24070). *S. pneumoniae* was incubated in 5% $CO_2$. Single colonies were suspended in 5 mL sterile saline, the turbidity was adjusted to 0.20 (Siemens MicroScan turbidity metre; equal to ~$1.0 \times 10^8$ colony forming units (CFU)/mL) and then diluted 200x in MHIIB (Mueller-Hinton II Agar (BD-211438)) medium. The MHIIB media was supplemented with 5% Lysed Horse Blood (Shanghai YuanMu Biological Technology Co. Ltd. YM-U161) for *S. pneumoniae*.

ACX compounds were dissolved in pure DMSO (Sigma 276855) to a concentration of 20 mg/mL. From this, twofold serial dilutions (from 6.4 to 0.00625 µg/mL) in DMSO were transferred to a 96-well source-plate (Axygen-wipp02280) with pure DMSO as a control. 2 µL from the source plate was then transferred into 96-well culture plates (Costar 3788) and inoculated with 198 µL of the bacterial suspension so that each well contained ~$5.0 \times 10^5$ CFU/mL bacteria, 1% DMSO and serially diluted compounds at 0 µg/mL and 0.0625-64 µg/mL in 100 µl medium. For NRS384, which was tested in a separate experiment, 99 µL of bacterial suspension was added to 1 µL of compound. The plates were incubated in ambient atmosphere at 37 °C for 20 h. The MIC values were determined by visual inspection as the lowest compound concentration that completely or significantly inhibits the growth of bacteria in the test medium.

### Generation of strains with reduced susceptibility

Strains with reduced susceptibility to ACX compounds were generated by WuXi Apptec on behalf of Acurx Pharmaceuticals, Inc., using *E. faecium* ATCC 700221 as parental strain.

Agar MICs for the wild-type strain were determined on Blood agar (Huankai). Compounds and linezolid were dissolved in 100% DMSO and vancomycin in ultrapure water. Strains were inoculated onto agar plates, incubated at 37 °C for 20-24 h after which colonies were suspended in sterile 0.9% saline at different bacterial concentrations. 5 µL of bacterial suspensions was individually spotted on agar plates supplemented with compounds at different concentrations, resulting in

final concentrations of $10^{-1}$, $10^{-2}$, $10^{-3}$, $10^{-4}$, $10^{-5}$, $10^{-6}$, and $10^{-7}$ per spot. After incubation for 20-24 h, bacterial growth was assessed by visual inspection. MIC was defined as the minimum compound concentration that significantly or completely prevented visible growth at an inoculum of $10^{-4}$ per spot.

For frequency of resistance (FOR) studies, colonies from the agar plate were suspended into sterile 0.9% saline to get $10^8$-$10^9$ CFU bacterial cells per mL. 100 µL of bacterial suspensions were spread on agar plates with serial dilutions of compound at 1x, 2x, 4x, 8x, and 16x MIC. The suspensions were also serially diluted with 0.9% saline and spotted onto agar to determine the actual inoculum. All agar plates were incubated at 37 °C for 48 h. FOR was defined as colony number on each compound-containing plate per CFU number of actual inoculum. Resistant colonies were subcultured on the corresponding compound plates to confirm that colonies were resistant and stable.

Confirmed resistant colonies were used for cross-resistance studies, as described above.

### Next-generation sequencing analysis

Paired-end short read sequencing data from selected mutants (generated by BGISEQ) as well as the corresponding wild-type parental strain was analysed against the reference genomes for *E. faecium* strain ATCC 700221 (https://genomes.atcc.org/genomes/f9d13a716ccb4f78) using Geneious R10.2.6 (Biomatters Ltd). Variants were called using the Find Variants/SNPs function with a minimum coverage of 10 and a minimum frequency of 0.8. Variants identified in the parental strain as well as the strains with reduced susceptibility were ignored, as presence in the parental strain indicates that these are not related to exposure to the antimicrobial. This identified the following mutations in the *polC* gene: D1103Y in strain Efa4; F1276L in strains Efa030, Efa038, and Efa039; F1276I in strains Efa1 and Efa037; F1276S in strain Efa3; A1281T in strains Efa036, Efa040 and Efa041. Strains Efa036, Efa039, and Efa040 each had only one polymorphism identified.

### Protein expression and purification

The genes for PolC (locus_tag: MIBBHLJO_01574) and DnaE (locus_tag: MIBBHLJO_00911) from *E. faecium* ATCC 700221 were codon-optimized for expression in *E. coli* and synthesized in vector pET28b by Twist Bioscience (San Francisco, CA). Expression constructs for variant PolC proteins were created with QuikChange Site-Directed Mutagenesis (Agilent) with oligonucleotides listed in Supplementary Table 3. F1276L, F1276S, and A1281T were introduced to assess the effect of these mutations on inhibition by ACX compounds. Mutations D972A and D974A were introduced to inactivate the polymerase activity and mutations D431A and E433A were introduced to inactivate the exonuclease activity (Fig. 1d, Supplementary Fig. 1).

Expression plasmids were transformed into *E. coli* Rosetta (DE3) pLysS cells and grown in LB medium with 35 mg/L chloramphenicol and 50 mg/L kanamycin at 37 °C to an optical density at 600 nm of 0.6–0.8 after which cultures were cooled to 16 °C as described before[53] and supplemented with 1 mM IPTG. After 16 h, cells were harvested by centrifugation (20 min at $4000 \times g$) and stored at −80 °C or directly lysed. The crude pellets were resuspended in a 5x volume (mL) to pellet mass (g) ratio in Buffer A (50 mM HEPES pH 8.0, 350 mM NaCl, 2 mM DTT) with 20 mM imidazole and then lysed by sonication. The lysate was centrifuged at 24,000 x *g* for 40 min, followed by a brief sonication to shear remaining DNA.

The N-terminally His-tagged *E. faecium* proteins were purified on a Ni-NTA column, followed by Q column chromatography as follows. The lysates were loaded onto a pre-equilibrated HisTrap FF column (5 mL column for every 5 g of original crude pellet) and eluted with a gradient of Buffer A with 500 mM imidazole. Peak fractions were pooled and diluted with Buffer B (50 mM HEPES pH 8.0, 2 mM DTT) to a NaCl concentration of no lower than 200 mM, to avoid precipitation. Pooled fractions were loaded onto a 1 mL Q HP column

pre-equilibrated to 200 mM NaCl with 80% Buffer B and 20% Buffer B2 (B with 1 M NaCl) and eluted with a gradient from 200 to 1000 mM NaCl (20-100% buffer B2). The concentration was determined for the peak fraction(s), snap-frozen in liquid nitrogen and stored at −80 °C. All PolC proteins yielded similar amounts of purified protein, but proteins with inactivated exonuclease demonstrated higher purity and were used for subsequent analyses.

DnaE from *E. coli* was purified as described before[33,34].

## Gel-based polymerase and exonuclease assays

A DNA substrate with an 11-nt single-stranded overhang was created by annealing a 37-nt template (Temp-Phospho-NoMis) and a 26-nt 5' carboxyfluorescein (6-FAM)-labelled oligonucleotide (61: 37/26 C T Phospho) pair at a 1.1:1 molar ratio. To test exonuclease activity, a DNA substrate was created by annealing a template (Temp-Phospho-NoMis) and a 6-FAM-labelled-oligonucleotide (oMU029) as above. Primer extension and exonuclease activity assays were performed at room temperature in Reaction Buffer (50 mM HEPES pH 8.0, 50 mM NaCl, 7.5 mM MgCl$_2$, 2 mM DTT) with 0.5 mg/mL BSA, 25 μM of each dNTP, 70 nM DNA substrate and 50 nM polymerase. Reactions were stopped in 35 mM EDTA and 65% formamide after 30 min, and subsequently separated on a denaturing 20% SequaGel for ~80 min at 30 W. The gel was imaged with a Typhoon Imager (GE Healthcare).

## Real-time DNA primer extension assay

Polymerase activity was monitored in real-time[31,32] with a 42-nt long 5' carboxyfluorescein-labelled DNA oligonucleotide template strand (DNASP8_42) annealed to a 30-nt primer oligonucleotide (oMU014). Polymerase activity in this assay is monitored by measuring fluorescence quenching of the labelled DNA template as a result of primer extension by PolC (for details see Supplementary Fig. 1b, c). Both primers were HPLC purified. Primer extension assays were performed in Reaction Buffer (see Gel-based polymerase and exonuclease assays) supplemented with 0.5 mg/mL BSA, 0.05% Tween20 and 25 uM of each dNTP (SRB) in 384-well Black Round Bottom plates (Corning 4514). The polymerase was diluted in the SRB and incubated for at least 5 min in the presence of inhibitor or solvent, with a final solvent concentration of 1%. The reaction was initiated by the addition of an equal volume of DNA substrate in SRB as described below. Fluorescence emission data was collected with a BMG Labtech PHERAstar FSX reader for 30 min at room temperature.

Compounds were dispensed with an Echo acoustic liquid handler (Beckman Coulter). All compounds were dissolved to 1 mM and 10 mM stocks in 100% DMSO and transferred to Echo LDV or PP source plates (Labcyte, Inc., CA), then dispensed to 384-well plates with a Labcyte Echo550 or Echo650 acoustic dispenser at volumes of 2.5 to 100 nL per well. Automated back-filling insured that all wells had a final concentration of 1% DMSO (total volume 100 nL) per 10 μL reaction. Each compound was tested in triplicate; a 'replicate range' consisted of 1.5-fold dilutions of inhibitor at a concentration from 2.5 μM to 100 μM (over 10 wells) or 225 μM (over 12 wells), a polymerase activity control in absence of inhibitor (only polymerase activity in 1% DMSO) and negative control (no polymerase added, measure for no activity). A Mantis microfluidic liquid dispenser (Formulatrix) was used to dispense 5 μL polymerase and 5 μL substrate DNA (individually diluted in SRB) per well. The final concentration of polymerase was 10 nM for PolC and adjusted for DnaE to 5 nM, to achieve rates comparable to PolC. The final concentration of DNA was 70 nM.

Data were analysed and standard error of the means (SEM) determined in Prism 8.4.2 and 10.2.3 (GraphPad). Raw time-course polymerase activity was plotted against time, with SEM of triplicates determined. The rate of activity (represented by signal quenching) was determined during the first 8 min (linear stage) by calculating the slope of fluorescence intensity change (converted to a positive value) over time (in seconds). The slopes were then normalized to percentage

activity (%); per 'replicate range', 100% activity was defined as polymerase activity in the absence of inhibitor (polymerase control), and 0% activity was determined in the absence of polymerase (DNA signal; negative control). The SEM was determined between 'replicate ranges'. The IC$_{50}$ was determined using the dose-response (inhibition) non-linear regression '[Inhibitor] vs. response (three parameters)' without constraints.

## Cryo-EM grid preparation and data acquisition

For structural studies, the DNA substrate was based on the same DNA sequence used for the real-time assays, but with a longer primer to ensure that the first complementary base on the template is a cytosine, thereby facilitating interaction with the inhibitors. Labelled (DNASP8_42) or non-labelled (oMU039) HPLC-purified template was annealed to a HPLC-purified primer (oMU017) at a 1:1 ratio, as previously.

1 μM purified PolC protein was incubated in the presence of 20 μM DNA substrate and 10 μM inhibitor (from a stock of ACX-801 or IBZ in 10% DMSO) for 20 min at room temperature in Reaction Buffer (50 mM HEPES pH 8.0, 50 mM NaCl, 7.5 mM MgCl$_2$, 2 mM DTT) with 0.05% Tween20. High concentrations of DNA were used to maximise DNA binding but lower concentrations of inhibitor to minimize aggregation and precipitates on grids. 3 μL was applied to copper Quantifoil R0.6/1 holey carbon grids, which were glow discharged for 45 s at 25 mA using a PELCO easiGlow™. The grids were then blotted (1 s at ~85% humidity and 20 °C), and flash frozen in liquid ethane using a Leica EM GP plunge freezer. Grid quality was assessed on a Talos Artica, then progressed for data collection to a Titan Krios.

Data was collected with Thermo Scientific EPU software in counting mode on a Titan Krios (FEI) electron microscope operating at 300 kV with a Gatan K3 detector and the slit width of the energy filter was 20 eV. Dose, magnification and effective pixel size are detailed in Supplementary Table 1.

## Model building and refinement

Collected data was processed in Relion 5.0[54], with workflows shown in Supplementary Figs. 2-3. The micrographs were corrected for motion using the MotionCor2[55], and then defocus was estimated with CTFFIND4 (4.1.10)[56]. For auto-picking of particles, Topaz (1.13.1.)[57] was used with a previously obtained model. The best classes from several rounds of 2D classification were selected for the generation of initial models. The best initial model was used as a reference model for 3D classification (2-3 classes); this generally produced two conformational states (apo state or polymerase state, without or with DNA substrate within the active site, respectively).

Classes with distinct DNA features within the polymerase domain were selected to determine ligand-bound structures. Classes without distinct DNA features within the polymerase domain were selected to determine ligand-free structures. Each selected class was 3D auto-refined with Blush regularization. This was followed by 2 rounds of Bayesian polishing and different rounds of CTF refinement. For the IBZ-bound map, an additional 3D classification round (after polishing and CTF refinement) was required to better separate particles without DNA bound to the polymerase domain. The final map was post-processed with a soft mask. A local resolution map was also generated to assist in model building in WinCoot (v0.9.8.7)[58].

Model building was performed using WinCoot and Phenix (v1.18.2-3874)[59] (Supplementary Table 1). The AlphaFold model (AF-A0A133CXW7-F1-v4) was aligned into the density map of the ACX-801-bound PolC using ChimeraX (1.8rc202405250712)[60] and the unmapped residues of the N-terminus were deleted. To fit the model, rigid body fitting was used in WinCoot and the structure from *G. kaustophilus* (PDB-3F2D) was also used as a reference point. The DNA molecule bound to the polymerase domain was generated in WinCoot according to the DNA substrate sequence, and 3 nucleotides of undetermined sequences were built into the density present in the

exonuclease domain. The inhibitors were positioned crudely using PyMOL (Molecular Graphics System, Version 3.0.3 Schrödinger, LLC) and further optimized in WinCoot with the corresponding CIF. The model was refined with real-space refinement and then followed several rounds of refinement with MolProbity as part of the Phenix package[61] and further adjustments in WinCoot.

Ligand interaction maps were drawn using LigPlot+ v2.2[62] and moe2024.0601 from the Chemical Computing Group[63,64] on the basis of PDB-9QPC (ACX-801) and 9QRL (IBZ).

### Construction and characterization of *C. difficile* strains

*E. coli* strains were cultured aerobically at 37 °C at 180 rpm in Lysogeny Broth (LB) (1% w/v tryptone (Oxoid), 1% w/v NaCl (J.T.Baker), 0.5% w/v yeast extract). *C. difficile* strains were cultured in Brain Heart Infusion broth (BHI) (3.7% w/v brain heart infusion broth (Oxoid), 0.5% w/v yeast extract). Culturing of *C. difficile* was carried out anaerobically at 37 °C at 140 rpm in an atmosphere of 85% $N_2$, 10% $CO_2$, and 5% $H_2$, using a Don Whitley VA1000 workstation. For solid media, 1.5% w/v agar (Alfa Aesar) was used. Where appropriate, media were supplemented with 50 mg/L colistin (Cayman Chemical), chloramphenicol (10 mg/L for liquid medium and 20 mg/L for agar plates), 30 mg/L kanamycin, and 15 mg/L thiamphenicol.

DNA fragments were amplified by PCR from total genomic DNA of *C. difficile* strain 630Δ*erm*[40] with Q5 High-Fidelity DNA Polymerase (NEB), using oligonucleotides oNM-001/oNM-003 (amplifying the *polC* gene with its presumptive promoter). The oNM-001/oNM-003 PCR fragment was digested using KpnI (Roche) and BamHI (Thermo Scientific). Digesting of plasmid pAP24[65] with enzymes allowed for the isolation of plasmid backbone after agarose gel electrophoresis. The backbone was ligated to the digested PCR fragments and transformed into chemically competent DH5α *E. coli* cells by heat shock; transformants were selected on media containing chloramphenicol. To generate plasmids carrying variant *polC* genes, mutations were introduced in the plasmids harbouring wild-type *polC* using QuikChange Site-Directed Mutagenesis and verified by Sanger sequencing using primers listed in Supplementary Table 3. An overview of all constructed plasmids and strains can be found in Supplementary Table 4 and 5.

*E. coli* donor strain CA434 was transformed with the plasmid to be transferred and subsequently grown on agar plates containing chloramphenicol and kanamycin. A 1 mL aliquot of 20 h grown liquid *E. coli* transformant culture was pelleted by centrifugation (2 min at 10,000 x *g*) and cells were resuspended in 200 µL non-sporulating *C. difficile* 630Δ*erm* culture. The mixture was plated onto non-selective BHI agar in 5 µL spots and after 20 h incubation, cells were harvested in 1 mL pre-reduced phosphate-buffered saline and plated in 100 µL aliquots on BHI agar supplemented with thiamphenicol and colistin. Putative transconjugants were confirmed by PCRs with primers oWKS-1070 and oWKS-1071 (targeting chromosomal *gluD*) and oWKS-1387, oWKS-1388, oWKS-1389, and oWKS-1390 (targeting *traJ* or *repA* on the plasmid). Note that these strains are merodiploid and carry a wild-type chromosomal copy of *polC*.

Susceptibilities of the parental and plasmid-carrying strains to IBZ and ACX-801 were determined using a simplified agar dilution method. In short, *C. difficile* strains were cultured for 20 h in liquid BHI. Bacterial cells were harvested by 10-min centrifugation at 3220 x *g* and resuspended in pre-reduced phosphate-buffered saline to a McFarland turbidity standard of 1.0. Suspensions were inoculated as 5 µL spots onto BHI agar containing different concentrations of inhibitor (0, 2, 4, 8, 16, 32, 64 mg/L). MICs (defined as the first dilution with severely impaired or no visible growth) were determined after 48 h and imaged on a SCAN500 plate imager (Interscience).

### Reporting summary

Further information on research design is available in the Nature Portfolio Reporting Summary linked to this article.

## Data availability

The cryo-EM density maps have been deposited in the Electron Microscopy Data Bank (EMDB) under accession codes EMD-53270 (PolC/ACX-801 complex); EMD-53320 (apo-PolC from the ACX-801 dataset), EMD-53319 (PolC/IBZ complex). The atomic coordinates have been deposited in the PDB with accession codes 9QPC (PolC/ACX-801 complex), 9QRN (apo-PolC from the ACX-801 dataset), and 9QRL (PolC/IBZ complex). Raw sequence data has been deposited in Gen-Bank under accession code PRJNA1190376. Source data for Fig. 1, Fig. 4, Supplementary Fig. 1 and Supplementary Table 2 are available with this paper as a Source Data file. Source data are provided with this paper.

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

## Acknowledgements

We thank Jan d'Engelbronner for strain constructions and Igor Sidorov for submission of the sequence data to GenBank. We thank staff members at the Netherlands Centre for Electron Nanoscopy (NeCEN) for help with data collection. Access to NeCEN was supported by the Netherlands Electron Microscopy Infrastructure (NEMI), project 184.034.014 of the National Roadmap for Large-Scale Research Infrastructure of the Dutch Research Council (NWO), to M.H.L. The collaboration projects POLSTOP2 and POLSTOP4MDRO are co-funded by the PPP allowance made available by Health~Holland, Top Sector Life Sciences & Health, to stimulate public-private partnerships, to W.K.S.

## Author contributions

M.U., A.H.F., N.M. and W.K.S. undertook genetic and biochemical studies, including protein purification and biochemical assays. M.U. and M.H.L. performed cryo-EM data collection, data processing and structure refinement. C.J.S. and M.R.B. contributed to the structure analyses. M.H.S., L.I.M., X.Y. and R.J.D. provided PolC inhibitors, contributed sequence data for reduced susceptible *E. faecium* isolates and provided frequency of resistance and susceptibility data. M.H.L. and W.K.S. directed the research. M.U., A.H.F., M.H.L. and W.K.S. wrote the manuscript with input from other authors.

## Competing interests

M.U., A.H.F., M.H.L., W.K.S. declare receiving funding from Acurx Pharmaceuticals, Inc for work described in this manuscript. M.H.S., C.J.S., M.R.B., L.I.M. and X.Y. are consultants to Acurx Pharmaceuticals, Inc. R.J.D. is an employee of Acurx Pharmaceuticals, Inc. M.H.S, L.I.M. and X.Y. are shareholders of Acurx Pharmaceuticals, Inc. N.M. declares no competing interests.
