## [Transparent Peer Review file · Nature Communications]

A unique inhibitor conformation selectively targets the DNA polymerase PolC of Gram-positive priority pathogens

Corresponding Author: Dr Wiep Klaas Smits

Version 0:

Reviewer comments:

Reviewer #1

(Remarks to the Author)

There is an urgent need to develop antimicrobials with novel modes of action to address the growing global threat of antimicrobial resistance in Gram-positive pathogens. Chemically synthesized PolC inhibitors represent a promising new class of compounds for treating infections caused by priority Gram-positive bacteria. A major advantage of these inhibitors lies in structural divergence that exist between the bacterial and the eukaryotic polymerases, reducing the risk of off-target effects.

In this study, Mia Urem and colleagues present cryoEM structures of *E. faecium* PolC bound to DNA, in complex with two inhibitors: IBZ, an orally administered, non-absorbed drug currently in clinical development for *Clostridioides difficile* infection, and ACX-801, a novel inhibitor with improved bioavailability. Both compounds form base-pairing interactions with the DNA at the active site, directly competing with incoming dGTP nucleotides. The nucleobase of each inhibitor occupies the same position as the incoming dGTP base observed in the *G. kaustophilus* PolC structure, thereby providing a structural rationale to their mechanism of action: blocking access to the active site and preventing nucleotide incorporation.

Both IBZ and ACX-801 adopt a non-planar conformation in which the aromatic R1 group, that is 3,4-dichlorobenzyl in IBZ and (2,2-difluoro-1,3-benzodioxol-5-yl)methyl in ACX-801, is inserted into a shallow hydrophobic pocket composed of four aromatic residues. This interaction likely contributes to the selectivity of these inhibitors for members of the Bacillota phylum over other bacterial groups, and their poor activity against DnaE-type polymerases *in vitro*. Among these residues, a key phenylalanine has been identified as an essential component of the active site for the inhibitory activity of this class of compounds.

From a structural biologist perspective, I find this work to be of very high quality.

The manuscript is written in a clear and concise manner, and the figures are informative. The results presented in this study are likely to provide important insights for the design of novel therapeutic molecules against high-priority pathogens by providing an atomic-level understanding of inhibitor binding.

I have only a few minor recommendations:

- Model building: The number of modelled residues varies by more than a hundred between the ACX-801 complex and the IBZ complex, suggesting a difference in stability (or homogeneity) in one or more regions of the protein between these two complexes. I suggest that the authors provide more detail on this point in the Methods section. Which regions of the protein are involved? Could it be ruled out that the binding of one inhibitor versus another affects the stabilization of these regions in the polymerase?
- Water molecules: Two water molecules are shown in the active site of PolC in complex with the ACX-801 inhibitor. How many water molecules were modelled in total? This is not specified in Extended Table 1 but should be included.
- Figure 4b, 4d – Figure 1e, 1f: In general, the data points on the dose-response curves are difficult to distinguish. I recommend that the authors use different symbols or a clearer colour scheme to better differentiate the curves corresponding to the different enzyme variants.
- Methods section: “Gel-based polymerase and exonuclease assays” – In the second sentence, the word “labelled” is unnecessarily repeated and should be revised.

Reviewer #2

(Remarks to the Author)

In this manuscript, the authors describe three structures of *Enterococcus faecium* PolC. Two structures show bound to DNA and inhibitors (ibezapolstat, IBZ, and ACX-801) that compete with dGTP for base-pairing to cytosine in the template DNA, and one structure is of the apo enzyme. PolC is the main replicative DNA polymerase in low GC-content Gram-positive bacteria, which includes *E. faecium*, *C. difficile*, *S. aureus*, and other human pathogens; *E. faecium* PolC is only the second PolC enzyme with experimentally determined structures. IBZ has completed phase 2 clinical trials for the treatment of *C. difficile* and is the first-in-class to reach this stage of development. The structures show that one moiety of the inhibitor binds to PolC in a pocket that is induced upon inhibitor binding. The pocket contains several aromatic residues, one of which (F1276 in VRE-PolC) is mutated in cells that acquire resistance to these and related inhibitors. This work is a timely and important contribution to the field and highlights the continuing necessity for experimental structure determination even in the era of AlphaFold.

Minor comments:

1. The authors highlight that the inhibitors adopt "unusual" non-planar conformations. It would be helpful to include a figure showing how the planar NMR structure (ref. 43) compares to the conformation observed in the PolC-bound structures.
2. Given the 2.8 to 4.5 Å resolution of the of the ACX-801 structure, the authors should show the density for the water molecules modeled in the structure of ACX-801 (Fig. 2e) and/or provide information on the b-factors for the water molecules and surrounding residues.
3. Using in vitro polymerase assays, the authors find that PolC enzyme containing F1276L and F1276S mutations have comparable activity to the wild-type enzyme (Fig. 4). Is F1276 in *E. faecium* PolC analogous to F1261 in *S. aureus* PolC? If so, how do the authors reconcile the data presented here to the data presented in reference #39, where the maximal polymerization rate of *S. aureus* PolC is ~10-fold lower when the enzyme contains an F1261L mutation? Is this a real difference in the effect of mutating this residue (i.e. the *S. aureus* enzyme is more sensitive to the mutation than is the *E. faecium* enzyme), or is the assay used here less sensitive to changes in polymerase activity? (Note: The description of the real-time DNA primer extension assay would benefit from a brief description of what is causing the decrease in fluorescence intensity that is the readout for polymerase activity in Fig. 4d.)
4. In the experiments where mutated versions of *C. difficile* PolC were expressed from plasmids in *C. difficile*, what the chromosomal copy of wild-type PolC present? How did the expression level of the plasmid-encoded PolC compare to that of the chromosomal copy?

Reviewer #3

(Remarks to the Author)

Antimicrobial resistance is a major global public health concern and bacterial replication proteins, due to their essential nature, can be potent targets for new antibiotics. The bacterial replicative DNA polymerases are structurally different from the corresponding human enzymes and have been identified as a target for novel antibiotics. Ibezapolstat is an antimicrobial that targets *C. difficile* PolC, the replicative polymerase of Gram positive bacteria. While ibezapolstat is effective in inhibiting PolC, the structural basis of this inhibition has not been delineated before.

In this study the authors used electron cryomicroscopy (cryoEM) to solve 2.8 to 3 Å structures of *E. faecium* PolC in complex with ibezapolstat or related novel inhibitor called ACX-801. They found that these small molecules adopt an altered conformation when bound to PolC compared to their solution structures. Moreover, by comparing the active site structures, the authors determined the source of selective binding of these compounds to the Gram Positive replicative polymerase PolC but not to the Gram Negative replicative polymerase DnaE.

The study is well performed and the conclusions are for the most part, supported by the observations. This work provides a strong foundation for further development of bacterial polymerase inhibitors.

A few relatively minor concerns for the authors to address:

Pg.6 "IC50 of 12.3 and 5.8 μM". It would be useful to provide a reference or a brief description for IC50.

Pg.6 "that the larger moiety of ACX-801 cannot be accommodated by the DnaE-type DNA polymerases." Figure 1 does not rule out alternative explanations. For instance, the altered charges on ACX-801 (eg. from the F on R3) might be responsible for this compound not binding to DnaE.

Pg.7 "and the nucleobase moiety of the inhibitor can stack onto the 3' terminal base ... (Fig. 2f)." Figure 2f does not show the stacking interaction.

Pg.8 "hydrogen bond with a water molecule... (Fig. 2d-e)." It would be informative to show the densities for the waters in a

supplemental figure along with the threshold value where they are visible.

Pg.8 "These R1 moieties are inserted...tyrosine1284" It will be useful to add a supplementary figure showing the densities of these critical residues.

Pg.9 "Based on the cryoEM structures, the observed mutations" From the phrasing it seems that the mutations were directly observed in the cryoEM structures. It would be good to clarify this.

Pg.10 "Both PolC variants...wild-type enzyme in our assay" If F1276 is analogous to F1261 of *S. aureus* PolC then the result presented in this manuscript is very different from the result reported in ref.39, where it was shown using transient state kinetics that F1261L mutant had an order of magnitude lower k_{pol} compared to the wild type enzyme. It would be useful to mention this difference in the discussion. Also, it would be useful to provide a brief description of the real-time DNA primer extension assay including how the signal is generated. While I am not very familiar with the method, it appears that this approach provides an overall rate of extension. If this is correct then it is important to establish that the differences in activity of the mutant and wildtype enzyme are arising from one or more steps of the catalytic cycle that is relevant to the *in vivo* situation and are not following a relatively inconsequential step like dissociation of the polymerase from the DNA substrate. A similar clarification would be useful for A1281T.

Pg.12 "as the structure of the PolC inhibitor in solid state (determined by NMR) is planer" The difference in the structures of the inhibitors in free and bound states is a major finding and a figure showing the two states would be informative.

Pg.20 "MgCl₂" 2 would be subscript.

Reviewer #4

(Remarks to the Author)

Version 1:

Reviewer comments:

Reviewer #1

(Remarks to the Author)

The authors have adequately addressed all of my comments. I can only congratulate them on their excellent work.

Reviewer #2

(Remarks to the Author)

The authors have adequately addressed my comments.

Reviewer #3

(Remarks to the Author)

The authors have satisfactorily addressed all our concerns/comments.

Reviewer #4

(Remarks to the Author)

RESPONSE TO REVIEWER COMMENTS

We appreciate the reviewers' positive and constructive feedback and addressed the minor points raised by all three reviewers as specified below.

Reviewer #1 (Remarks to the Author):

There is an urgent need to develop antimicrobials with novel modes of action to address the growing global threat of antimicrobial resistance in Gram-positive pathogens. Chemically synthesized PolC inhibitors represent a promising new class of compounds for treating infections caused by priority Gram-positive bacteria. A major advantage of these inhibitors lies in structural divergence that exist between the bacterial and the eukaryotic polymerases, reducing the risk of off-target effects.

In this study, Mia Urem and colleagues present cryoEM structures of *E. faecium* PolC bound to DNA, in complex with two inhibitors: IBZ, an orally administered, non-absorbed drug currently in clinical development for *Clostridioides difficile* infection, and ACX-801, a novel inhibitor with improved bioavailability. Both compounds form base-pairing interactions with the DNA at the active site, directly competing with incoming dGTP nucleotides. The nucleobase of each inhibitor occupies the same position as the incoming dGTP base observed in the *G. kaustophilus* PolC structure, thereby providing a structural rationale to their mechanism of action: blocking access to the active site and preventing nucleotide incorporation.

Both IBZ and ACX-801 adopt a non-planar conformation in which the aromatic R1 group, that is 3,4-dichlorobenzyl in IBZ and (2,2-difluoro-1,3-benzodioxol-5-yl)methyl in ACX-801, is inserted into a shallow hydrophobic pocket composed of four aromatic residues. This interaction likely contributes to the selectivity of these inhibitors for members of the Bacillota phylum over other bacterial groups, and their poor activity against DnaE-type polymerases in vitro. Among these residues, a key phenylalanine has been identified as an essential component of the active site for the inhibitory activity of this class of compounds.

From a structural biologist perspective, I find this work to be of very high quality.

The manuscript is written in a clear and concise manner, and the figures are informative. The results presented in this study are likely to provide important insights for the design of novel therapeutic molecules against high-priority pathogens by providing an atomic-level understanding of inhibitor binding.

I have only a few minor recommendations:

- Model building: The number of modelled residues varies by more than a hundred between the ACX-801 complex and the IBZ complex, suggesting a difference in stability (or homogeneity) in one or more regions of the protein between these two complexes. I suggest that the authors provide more detail on this point in the Methods section. Which regions of the protein are involved? Could it be ruled out that the binding of one inhibitor versus another affects the stabilization of these regions in the polymerase?

The structures of 801-bound PolC and IBZ-bound PolC differ in resolution: 2.8 Å and 3.2 Å, respectively. This is most visible in the OB domain and exonuclease domain that are located at the periphery and loosely associated with the rest of the polymerase (compare Sup Fig 2h and Sup Fig 3h). Due to the lower map resolution in this region, we could build only 15 residues of the OB domain in the IBZ-bound molecule, compared to 99 residues of the OB domain in the ACX-801 bound structure. In the exonuclease domain several loops (33 residues) were not modelled in the IBZ-bound structure, whereas only two residues were omitted in the exonuclease domain of the 801-bound structure.

We have now included in the text:

"Due to the lower resolution of the IBZ-bound structure only 15 residues of the OB domain (consisting of 110 residues) and 175 residues of the exonuclease domain (consisting of 211 residues) could be modelled in this map. Both domains are located at the periphery of the polymerase. In comparison, in the higher-resolution 801-bound structure, only 12 residues were omitted in the OB domain and 2 residues in the exonuclease domain."

- Water molecules: Two water molecules are shown in the active site of PolC in complex with the ACX-801 inhibitor. How many water molecules were modelled in total? This is not specified in Extended Table 1 but should be included.

This has been added to the Extended Table 1.

- Figure 4b, 4d – Figure 1e, 1f: In general, the data points on the dose-response curves are difficult to distinguish. I recommend that the authors use different symbols or a clearer colour scheme to better differentiate the curves corresponding to the different enzyme variants.

We have adjusted the graphs to improve clarity.

Specifically:

- **Figure 1e, f: Per editorial request we now show individual datapoints. To improve clarity, we have increased symbol size and included different symbols per curve. The number of replicates are now indicated in the Figure legend.**
- **Figure 4b; We have increased symbol size, and now use filled symbols to increase clarity.**
- **Figure 4d; To improve clarity, we have increased symbol size and included different symbols per curve. As a result, the symbols fully obscured the shading from the original figure, so we chose to omit this from the revised version. The legend now clearly indicates that the datapoint represents a mean, with the error omitted as it is obscured by the size of the symbols.**

- Methods section: “Gel-based polymerase and exonuclease assays” – In the second sentence, the word “labelled” is unnecessarily repeated and should be revised.

The text was rephrased as “A DNA substrate of 37 bp with a single-stranded overhang was created by annealing template (Temp-Phospho-NoMis) and 5' carboxyfluorescein (6-FAM)-labeled oligonucleotide (61:37/26 C T Phospho) pair at a 1.1:1 molar ratio.”.

Reviewer #2 (Remarks to the Author):

In this manuscript, the authors describe three structures of *Enterococcus faecium* PolC. Two structures show bound to DNA and inhibitors (ibezapolstat, IBZ, and ACX-801) that compete with dGTP for base-pairing to cytosine in the template DNA, and one structure is of the apo enzyme. PolC is the main replicative DNA polymerase in low GC-content Gram-positive bacteria, which includes *E. faecium*, *C. difficile*, *S. aureus*, and other human pathogens; *E. faecium* PolC is only the second PolC enzyme with experimentally determined structures. IBZ has completed phase 2 clinical trials for the treatment of *C. difficile* and is the first-in-class to reach this stage of development. The structures show that one moiety of the inhibitor binds to PolC in a pocket that is induced upon inhibitor binding. The pocket contains several aromatic residues, one of which (F1276 in VRE-PolC) is mutated in cells that acquire resistance to these and related inhibitors. This work is a timely and important contribution to the field and highlights the continuing necessity for experimental structure determination even in the era of AlphaFold.

Minor comments:

1. The authors highlight that the inhibitors adopt "unusual" non-planar conformations. It would be helpful to include a figure showing how the planar NMR structure (ref. 43) compares to the conformation observed in the PolC-bound structures.

We thank the reviewer for highlighting this point. In the revised version of the manuscript we further corrected and clarified this aspect. We have qualified the statement by adding "uracil-based" to more clearly indicate that these are historic data obtained with a previous generation of PolC inhibitors and removed the reference to NMR as the solid state structures were obtained by X-ray crystallography. To underscore that the next generation compounds (like ACX-801) are likely to adopt a similar conformation, we performed a conformational search for the lowest energy conformations for ACX-801 (Supplemental Fig. 7). The 10 lowest energy conformations all show the same near-planar extended conformation that is clearly distinct from the conformation of ACX-801 in complex with PolC and DNA. The dihedral of the lowest energy conformation shows an angle of 172.1° (compared to -88.8° observed in our structure).

2. Given the 2.8 to 4.5 Å resolution of the of the ACX-801 structure, the authors should show the density for the water molecules modeled in the structure of ACX-801 (Fig. 2e) and/or provide information on the b-factors for the water molecules and surrounding residues.

We have introduced a panel illustrating this in the revised version of the manuscript (Supplementary Fig. 2e and accompanying legend). We have furthermore added to the legend of Figure 2h: "The water molecules and surrounding residues are well defined in the cryo-EM map, with B-factors ranging between 15-30 Å², which is lower than the average B-factor of 47 Å² for the entire polymerase".

3. Using in vitro polymerase assays, the authors find that PolC enzyme containing F1276L and F1276S mutations have comparable activity to the wild-type enzyme (Fig. 4). Is F1276 in *E. faecium* PolC analogous to F1261 in *S. aureus* PolC? If so, how do the authors reconcile the data presented here to the data presented in reference #39, where the maximal polymerization rate of *S. aureus* PolC is ~10-fold lower when the enzyme contains an F1261L mutation? Is this a real difference in the effect of mutating this residue (i.e. the *S. aureus* enzyme is more sensitive to the mutation than is the *E. faecium* enzyme), or is the assay used here less sensitive to changes in polymerase activity? (Note: The description of the real-time DNA primer extension assay would benefit from a brief description of what is causing the decrease in fluorescence intensity that is the readout for polymerase activity in Fig. 4d.)

The reviewer is correct in assuming that the F1276 in *E. faecium* corresponds to F1261 in the *S. aureus* PolC (see also Figure 3b that contains the actual alignment). There are, however, key differences between our study and ref #39:

- Our study uses heterologously produced and purified PolC of *E. faecium*; our purification differs from that reported in reference #39 that might affect protein activity and stability, and additionally we cannot exclude that species-specific differences exist, despite the conserved active site of PolC.
- The mutations introduced to inactivate exonuclease activity are different between ref #39 and our study; it is unknown how this could potentially affect results
- Ref #39 includes beta-clamp protein in the assays, whereas our assays only employ polymerase protein.
- Ref #39 assesses pre-steady-state kinetics, whereas our assay looks at overall (steady state) polymerase activity (as pointed out by Reviewer 3 also), and for that reason cannot be directly compared. We note however, that the gel-based assay depicted in Fig. 2 of Ref #39 (provided below), that is perhaps in set-up most similar to our assays, appears to show similar polymerase activity for PolC-WT and PolC-F1261L (compare the red boxed areas). This is in line with our observations.

We have added a statement to acknowledge that our result appear to differ from those previously reported: “We note that in *S. aureus* the equivalent protein (PolCF1261L) showed a reduced rate of nucleotide incorporation in pre-steady-state kinetic experiments in the presence of clamp protein, but showed comparable rates to wild type in a steady-state gel-based polymerase assay.”). As we cannot conclusively contribute the difference to one specific aspect, we prefer not to speculate on this further in the revised manuscript.

With respect to the details of our polymerase assay: our initial submission did not include experimental details as the assay is extensively described in ref 31 and 32, cited in the Methods section of our manuscript. We acknowledge however that it may help the reader to have some more information. A graphical depiction of the assay is now included as Supplemental Fig. 1b and Supplementary Fig. 1c and we have added a comprehensive statement to the Methods section. Reference to the new Supplemental Figure has also been added to the legend of Fig. 1e and Fig. 4b.

4. In the experiments where mutated versions of *C. difficile* PolC were expressed from plasmids in *C. difficile*, what the chromosomal copy of wild-type PolC present? How did the expression level of the plasmid-encoded PolC compare to that of the chromosomal copy?

Yes, the reviewer is correct that the wild type copy of the *polC* gene is present in these strains, similar to experiments published for *B. subtilis* (Ott 1986 J Bacteriol, doi:10.1128/jb.165.3.951-957.1986). We have added a statement to this effect to the Methods in the revised version of the manuscript. We did not assess expression levels of plasmid-expressed PolC levels as the proteins are indistinguishable from chromosomally expressed PolC.

We have, however, shown previously that the copy number of the shuttle vector used in our experiments is approximately 4x the chromosomal copy number (see Figure 7a in <https://www.nature.com/articles/s41467-020-14382-1>), provided below for your reference (IB30 is a strain that contains a plasmid with the same replicon).

We do not consider it likely plasmid copy number affects the interpretation of our results as

1. We expect the mutant alleles to be dominant over the wild type allele (that is: the mutant allele has reduced binding of the compound and polymerase activity is retained in the presence of the compounds, compared to wild type)
2. Our unpublished data indicates that (very) low level expression of the mutant allele is sufficient to confer resistance, induction of wild type PolC does not lead to resistance against IBZ, and overexpression of the mutant PolC does not lead to appreciable changes in growth on solid media. This data is shown below, for review: Ptet is an anhydrotetracyclin (ATc)-inducible promoter that shows leaky expression (i.e. very low level) in the absence of inducer (ATc -).

Reviewer #3 (Remarks to the Author):

Antimicrobial resistance is a major global public health concern and bacterial replication proteins, due to their essential nature, can be potent targets for new antibiotics. The bacterial replicative DNA polymerases are structurally different from the corresponding human enzymes and have been identified as a target for novel antibiotics. Ibezapolstat is an antimicrobial that targets *C. difficile* PolC, the replicative polymerase of Gram positive bacteria. While ibezapolstat is effective in inhibiting PolC, the structural basis of this inhibition has not been delineated before.

In this study the authors used electron cryomicroscopy (cryoEM) to solve 2.8 to 3 Å structures of *E. faecium* PolC in complex with ibezapolstat or related novel inhibitor called ACX-801. They found that these small molecules adopt an altered conformation when bound to PolC compared to their solution structures. Moreover, by comparing the active site structures, the authors determined the source of selective binding of these compounds to the Gram Positive replicative polymerase PolC but not to the Gram Negative replicative polymerase DnaE.

The study is well performed and the conclusions are for the most part, supported by the observations. This work provides a strong foundation for further development of bacterial polymerase inhibitors.

A few relatively minor concerns for the authors to address:

Pg.6 "IC₅₀ of 12.3 and 5.8 μM". It would be useful to provide a reference or a brief description for IC₅₀.

We have now defined IC₅₀ (inhibitory concentration at which activity is 50%) at first use on page 6.

Pg.6 "that the larger moiety of ACX-801 cannot be accommodated by the DnaE-type DNA polymerases." Figure 1 does not rule out alternative explanations. For instance, the altered charges on ACX-801 (eg. from the F on R3) might be responsible for this compound not binding to DnaE.

We agree and have adjusted the text ("This suggests that the larger moiety of ACX-801 cannot be accommodated by DnaE-type DNA polymerases, though we cannot rule out that other factors contribute to the reduced inhibition").

Pg.7 "and the nucleobase moiety of the inhibitor can stack onto the 3' terminal base ... (Fig. 2f)." Figure 2f does not show the stacking interaction.

We appreciate the reviewer pointing out that the statement was incorrectly referenced. We have rephrased to "the inhibitor can stack onto the 3' terminal base of the primer strand (Supplemental Fig. 2e and 3e) while base-pairing with the dCMP in the template strand (Fig. 2f)."

Pg.8 "hydrogen bond with a water molecule...(Fig. 2d-e)." It would be informative to show the densities for the waters in a supplemental figure along with the threshold value where they are visible.

A new panel (Supplemental Fig. 2e) has been added to show the density of the water molecules.

Pg.8 "These R1 moieties are inserted...tyrosine1284" It will be useful to add a supplementary figure showing the densities of these critical residues.

These data are now included as Supplemental Figs. 2f and 3f.

Pg.9 "Based on the cryoEM structures, the observed mutations" From the phrasing it seems that the mutations were directly observed in the cryoEM structures. It would be good to clarify this.

We agree and have clarified the text as follows: "Based on the cryo-EM structures, the three mutations (F1276I, F1276L and F1276S) that were observed in the resistance selection experiments likely lead to loss of

the interactions with both the N^2 moiety and the nucleobase (Fig. 4a, Fig. 2d-e, Fig. g-h, Supplementary Fig. 6)."

Pg.10 "Both PolC variants...wild-type enzyme in our assay" If F1276 is analogous to F1261 of *S. aureus* PolC then the result presented in this manuscript is very different from the result reported in ref.39, where it was shown using transient state kinetics that F1261L mutant had an order of magnitude lower k_{pol} compared to the wild type enzyme. It would be useful to mention this difference in the discussion. Also, it would be useful to provide a brief description of the real-time DNA primer extension assay including how the signal is generated. While I am not very familiar with the method, it appears that this approach provides an overall rate of extension. If this is correct then it is important to establish that the differences in activity of the mutant and wildtype enzyme are arising from one or more steps of the catalytic cycle that is relevant to the *in vivo* situation and are not following a relatively inconsequential step like dissociation of the polymerase from the DNA substrate. A similar clarification would be useful for A1281T.

We kindly refer to our response to point 3 of Reviewer 2 for this point.

Pg.12 "as the structure of the PolC inhibitor in solid state (determined by NMR) is planer" The difference in the structures of the inhibitors in free and bound states is a major finding and a figure showing the two states would be informative.

We kindly refer to our response to point 1 of Reviewer 2 for this point.

Pg.20 "MgCl₂"² would be subscript.

Corrected as suggested.